# Development of a Flex-Seq SNP panel for raspberry (*Rubus idaeus* L.) and validation through linkage map construction and identification of QTL for several traits of agronomic importance to raspberry breeding

Jahn Davik[1]*, Paolo Zucchi[2], Matteo Buti[3], Linda Milne[4], Julie Graham[5], Daniel James Sargent[6]

1 Department of Molecular Plant Biology, Norwegian Institute of Bioeconomy Research, Ås, Norway, 2 Sant'Orsola SCA, Department of Fruit Production, Pergine Valsugana, Trento, Italy, 3 Department of Agriculture, Food, Environment and Forestry (DAGRI), University of Florence, Florence, Italy, 4 Informational and Computational Sciences, The James Hutton Institute, Dundee, United Kingdom, 5 Cell and Molecular Sciences, The James Hutton Institute, Dundee, United Kingdom, 6 NIAB, Cambridge, United Kingdom

* jahn.davik@nibio.no

## Abstract

High-throughput and reproducible genotyping platforms are critical for advancing genetic research and breeding in horticultural crops. Here, the development and validation of a custom single nucleotide polymorphism (SNP) panel using the Flex-Seq genotyping platform for red raspberry (*Rubus idaeus* L.) is described. SNPs were derived from existing linkage maps and RNA-seq data, resulting in a panel of 5,639 high-confidence, bi-allelic markers distributed across the seven chromosomes of the *R. idaeus* 'Malling Jewel' reference genome. The panel was used to genotype 457 red raspberry accessions including 161 individuals from a bi-parental mapping population (Paris×486), enabling the construction of high-density linkage maps and the identification of quantitative trait loci (QTL) for fruit size, leaf colour, plant vigour, and thorn density. Genome-wide association studies (GWAS) identified a major QTL for thornlessness on chromosome 4, co-locating with a candidate HOX3 gene, and multiple QTLs associated with anthocyanin biosynthesis genes for leaf colour. The SNP panel demonstrated utility for linkage mapping and trait association analyses, offering a powerful resource for marker-assisted selection and genetic improvement in red raspberry.

## Introduction

Red raspberry (*Rubus idaeus* L.) is the most economically important species in the highly diverse Rosaceous genus *Rubus*. The species is prized primarily for its edible

**Data availability statement:** *****A/PROS at Accept: Please follow up with the authors for data repository information available at Accept.***** All relevant data are within the manuscript and its supporting information files. In addition the sequence data are be uploaded to ArrayExpress and will as of now be released by the end of 2025 (ArrayExpress Accession number: E-MTAB-15650 Release date: 2025-12-31).

**Funding:** The research leading to these results has received funding from the European Union's Horizon 2020 research and innovation programme under grant agreement No. 101000747. The author is solely responsible for its content, it does not represent the opinion of the European Commission and the Commission is not responsible for any use that might be made of data appearing therein. https://research-and-innovation.ec.europa.eu/funding/funding-opportunities/funding-programmes-and-open-calls/horizon-2020_en This grant was awarded to JD, PZ, and LM. The funder did not play any role in any part of the research presented here.

**Competing interests:** The authors have declared that no competing interests exist.

drupelets that develop into sweet, aromatic, usually red 'berries' which detach easily from their receptacles and are predominantly sold for fresh consumption throughout the world. The global production of red raspberries has steadily increased over the past 20 years, in part due to the development of high-quality varieties that can be produced throughout the temperate regions of the world, and produce high yields, delivering profitable returns for growers and retailers. To assist in the development of new red raspberry varieties, commercial breeding programmes are turning to the use of molecular markers to assist the selection process. Recently, several genome sequences for red raspberry have been published for the species [1–4], facilitating the development of large numbers of markers for use in high-throughput genotyping.

Rapid and reliable genotyping is one of the key foundations of any genetics study, and can be used to study diversity, develop genetic linkage maps [5,6] and identify markers associated with traits of agronomic importance [7,8]. High quality genotyping has been achieved in many crop species through the use of hybridisation-based genotyping microarrays, such as the Illumina Infinium arrays developed in apple [9], cherry [10], and peach [11], and the Affymetrix Axiom arrays developed in rose [12], apple [13] and strawberry [14], which produce reliable genotypes for many thousands, to hundreds of thousands of SNPs. Traditionally however, genotyping platform development has been the rate-limiting step in advancing the use of high-throughput SNP genotyping for understanding of the genetic control of agronomically-important traits in many crop species, due to prohibitive development costs.

Genotyping-by-sequencing (GBS) [15] is a relatively inexpensive method for the identification of large numbers of polymorphic SNP loci within a genome. It exploits the distribution of restriction enzyme sites within a given genome to produce reduced-representation genomic libraries, and therefore requires no prior genomic information, using short-read sequencing to produce sequence data for SNP loci distributed genome-wide within any species. The technique has been used successfully in raspberry to develop SNP-based linkage maps of several mapping populations, either as the predominant marker type used, such as in the 'Heritage' × 'Tulameen' linkage map [16], or to 'saturate' linkage maps previously-produced using other marker types, such as in the case of the 'Glen Moy' × 'Latham' linkage map [17]. The 'Glen Moy' × 'Latham' linkage map has shown utility in several further studies permitting QTL for a range of traits to be identified, including fruit ripening [17], hyperspectral signatures correlated to several physical traits [18], crumbly fruit [19] and polyphenol metabolites [20]. However, there are many challenges with SNP genotyping using the GBS technique, including high levels of missing, or error-prone data, and most importantly, low reproducibility of genotyping between progenies.

Recently, several SNP genotyping platforms employing sequence-capture and high-throughput sequencing have been developed that are significantly more reproducible between experiments than GBS, but far more cost-effective and flexible than hybridisation-based genotyping arrays, and as such, are attractive for genetic investigation in crop species. These include the Flex-Seq platform, that has been successfully used to develop high density genotyping panels for blueberry [21,22], and the single primer enrichment technology (SPET) genotyping that has been used

to develop genotyping panels for tomato and aubergine [23] and lettuce [24]. Using these platforms, the cost of whole-genome SNP genotyping has reduced to the point whereby it is now cost-effective to employ the approach for almost any genetics study. Flex-Seq [25] is a SNP genotyping platform that employs sequence capture using targeted probes followed by Illumina short-read sequencing. The platform is claimed to provide highly accurate, reproducible data of a similar quality and consistency to data from genotyping arrays [25] and since the platform relies on hybridisation of oligo-nucleotide probes, new markers can be added to existing SNP panels far more readily than with array-based genotyping platforms. Flex-Seq has several advantages over GBS; it requires prior knowledge of genomic variation of interest, which means it is highly targeted and customisable, ensuring specific regions of interest, or overall genome coverage can be achieved with a given SNP panel. It is also more robust to variation in DNA quality and as such, gives a more reproducible and reliable output in terms of sequence depth and representation, as well as between experiments than can be achieved with GBS [25].

For these reasons, in this investigation, the Flex-seq SNP genotyping platform (LGC Genomics GmbH, Germany) was employed to develop a SNP genotyping panel for red raspberry (*R. idaeus*). Previously-published molecular markers that had been mapped to genetic maps of several red raspberry varieties [16,17] were used to create a framework SNP set spanning the 'Malling Jewel' reference genome sequence [4] and these were supplemented with robust SNPs character-ised from RNASeq data of developing fruit tissues previously reported by [3]. The SNP panel was used to characterise a red raspberry diversity panel and a bi-parental mapping population, and QTL for several traits were identified to demon-strate the efficacy of the panel for genetics studies in *R. idaeus*.

## Materials and methods

### Plant material and phenotyping

A total of 457 red raspberry genotypes comprising a diversity panel that included 27 red raspberry varieties (S1 File), 269 breeding selections from the Sant'Orsola raspberry breeding programme, and 161 progeny of an F1 mapping population were studied in the field experiments. The bi-parental $F_1$ mapping population was raised from a controlled cross between the Sant'Orsola raspberry selections 'Paris' (female) and '486' (male) (denoted Paris×486) was phenotyped in the same field trial. All raspberry germplasm was grown and maintained in substrate in a common breeding plot following standard cultivation practices at the Sant'Orsola S.C.A breeding centre at Altopiano della Vigolana, Trentino, Italy.

The field experiment followed an augmented design with regular control varieties. Subjective scores were taken regu-larly through each of the four experimental years using ordinal scales (0–10) for four horticultural traits; plant vigour, leaf colour, fruit size, and thorn density, the highest scores indicating more vigorous, darker leaves, larger and higher thorn density, respectively. Where necessary, data were transformed using Box-Cox transformation [26]. Each plot consisted of six clonal plants of each genotype. Phenotyping was performed over a total of four years and were analysed using ASReml-R [27] with entries as fixed and additional factors (e.g., years, plant cycle, and interaction terms) as random fac-tors in order to get best linear unbiased estimators (BLUEs). As input for the quantitative trait linkage analysis (QTL) and the genome wide association analysis (GWAA) BLUEs across all years were used.

### Marker development

**Selection of SNPs.** Data for SNP marker development was derived from two sources. Firstly, markers that had previously identified using the Genotyping by Sequencing protocol of [15] and mapped in the 'Heritage' × 'Tulameen' (H × T) mapping population [28] and the 'Glen Moy' × 'Latham' mapping population [17] were retrieved from those datasets. Sequence Tag data containing the mapped SNPs were aligned to the 'Malling Jewel' reference sequence [4] using Blastn [29] and the SNP positions within the reference sequence were determined.

SNPs were then identified from red raspberry RNASeq data derived from the raspberry cultivars 'Anitra', 'Glen Ample', 'Varnes' and 'Veten' previously published by [3]. The quality of the RNASeq libraries was assessed using FastQC v0.11.9 [30], and poor quality reads and TruSeq adapters sequences were filtered out with Trimmomatic 0.36 [31] (parameters: ILLUMINACLIP:TruSeq3-PE.fa:2:30:10 LEADING:5 TRAILING:5 SLIDINGWINDOW:4:15 MINLEN:100). Filtered reads of all the libraries were mapped to the 'Malling Jewel' genome sequence [4] using the STAR version 2.7.3a [32] aligner in a two-pass mode to improve splice junction detection. In the first pass, reads from each sample were mapped to a STAR-generated genome index incorporating known splice junctions from the provided GTF annotation file [32]. The detected splice junctions from the first-pass alignment of each sample were then used to build a new second-pass STAR genome index for that specific sample. The same sample was then realigned using this refined index to improve mapping accuracy at exon-exon junctions. Aligned reads were sorted and duplicates marked using GATK [33] MarkDuplicates v4.4.0.0. Given the spliced nature of RNASeq data, GATK SplitNCigarReads was applied to preprocess reads before variant calling. Single-nucleotide polymorphisms (SNPs) and small insertions/deletions (indels) were identified using GATK HaplotypeCaller in GVCF mode for each sample, followed by joint genotyping with GATK GenotypeGVCFs to produce a final multi-sample VCF file. The raw VCF file was then filtered using bcftools v1.21-25-g9d314180 [34] based on quality thresholds, including variant quality (QUAL > 30), mapping quality (MQ > 40), read depth (DP > 10), minor allele frequency (AF > 0.05), and BaseQRankSum (−2–2), to retain high-confidence variant calls for downstream analysis. The physical positions of the SNPs identified from each dataset were used to create a BED file that was submitted to the Flex-Seq panel design pipeline (LGC Genomics GmbH, Germany) for probe and adapter design, synthesis, and validation.

**SNP genotyping.** Approximately 50 mg of leaf material was harvested from each of the 457 genotypes, preserved using the BioArk leaf collection kit (LGC Genomics GmbH, Germany) and sent DNA extraction using the oKtopure DNA extraction platform (LGC Genomics GmbH, Germany) and sequence data were generated using the Flex-Seq SNP panel and sequencing pipeline using the NexSeq 500 Illumina sequencing platform at LGC Genomics (LGC Genomics GmbH, Germany). In order to capture data from all SNPs within the sequenced amplicons, the raw reads were trimmed using Trimmomatic 0.39 [31] using the following settings: LEADING:5 TRAILING:5 SLIDINGWINDOW:4:15 MINLEN:100, and were aligned using BWA-mem version 0.7.17-r1188 [35] to an indexed version of the 'Malling Jewel' raspberry reference genome [4]. The SNPs identified from the sequence data were called with the HaplotypeCaller in GATK [36]. Filtering was done with bcftools [34] keeping only biallelic SNPs, not accepting missing values below 0.1, requiring read quality to be above 30, retaining read depths below 300 and above 10, accepting BaseQRankSum below 0.3 and above −0.3, and finally, a minor allele frequency above 0.01. Following SNP identification, marker density was plotted from the genomic locations of SNPs in the 'Malling Jewel' reference genome [4] using the R package rMVP [37] to show the distribution of SNPs throughout the raspberry genome. A Neighbour Joining tree showing the relationships between the 457 genotypes used in this study was constructed using data from all polymorphic SNPs with BioPython using default parameters.

## Linkage mapping

The SNP genotypes called in the 161 segregating full-sib family were further scrutinised to identify out-crossing, contaminated samples and rogue genotypes. Using an in-house R-code, the marker data was interrogated and 'impossible genotypes' (genotypes that could not have been derived from the mating of the two parental genotypes, i.e., the presence of genotype BB in the cross AA×AB) were flagged for each progeny. To account for a limited amount of erroneous SNP calling from the sequence data, a threshold of more than five impossible genotypes per progeny was set and any progeny exceeding this value were removed from further analysis. Any remaining impossible genotypes in the retained progeny were set to missing values before linkage analysis, QTL mapping and GWAS were performed.

Joinmap 4.1 (Kyasma, NL) was used to group markers into linkage groups using the groupings tree function and a LOD threshold of 8.0. The genetic groupings were compared to the physical position of the markers on the raspberry 'Malling Jewel' reference genome and markers whose genetic linkage group did not match with their physical chromosome were

removed from further analysis. The remaining markers were ordered by physical position along each chromosome and marker data imputed according to the methodology of [16].

To demonstrate the applicability of the SNP panel to linkage mapping and QTL analysis, a subset of markers were selected from all heterozygous SNPs for map construction. A total of 25 markers heterozygous in the male parent and 25 heterozygous in the female parent that were approximately evenly distributed throughout each chromosome based on physical position were selected from each of the seven chromosomes of the *R idaeus* genome, along with a further 25 markers from each chromosome that were heterozygous in both parents. These 525 segregating markers were combined with the 20 most significant markers identified from a Kruskal-Wallis analysis for each of the four phenotypes investigated (65 markers in total as some significant markers were common between traits) to give a total of 590 markers. Regression mapping, implemented in JoinMap 4.1 (Kyasma, NL) using default parameters was performed using the 590 markers to calculate a linkage map for both the male and female parents of the Paris×486 mapping population. The resultant linkage maps were plotted using MapChart 2.0 [38] with lines connecting common markers heterozygous and segregating in both parental maps (i.e., markers with the genotype AB×AB). Colinearity between genetic and physical positions of the mapped markers was determined by producing scatter plots of the female and male marker distribution, plotted using ggplot2 [39].

**QTL analysis of traits segregating in the Paris×486 mapping progeny.** The best linear unbiased estimates calculated (BLUEs) for the mean phenotypic values calculated as described above were used as input for QTL analysis in the Paris×486 mapping population. A Kruskal-Wallis analysis was performed for each of the phenotypic trait datasets and the 20 most significant markers for each trait were included in the linkage map created using regression mapping. QTL analyses were performed using interval mapping implemented in MAPQTL 6.0 (Kyazma, NL) with a step size of 1.0 cM, from which percentage phenotypic variance explained and associated LOD values were calculated. A permutation test to determine the genome-wide LOD threshold for each trait was performed using 20,000 permutations. Significant QTL identified for each of the four traits were plotted on the Paris×486 parental linkage maps using MapChart 2.0 [38].

## Genome-wide association (GWAS) analyses

The best linear unbiased estimates (BLUEs) for plant vigour, leaf colour, fruit size, and thorn density calculated for the red raspberry accessions from the diversity panel that were both genotyped and phenotyped. The BLUEs were used as input for the GWAS analyses. A GWAS was performed using all polymorphic markers in the diversity panel. The GWAS was conducted using the single-locus mixed linear model [MLM, 40] and the multiloci Bayesian-information and linkage-disequilibrium iteratively nested keyway model [BLINK, 41] implemented in the GAPIT R Software package, version 3 [42]. To account for population structure and relatedness among entries, five principal components and the VanRaden [43] relationships matrix were included in the analyses. Manhattan plots showing the most significant associations on all seven red raspberry chromosomes were plotted using ggplot2 [39].

## Candidate gene identification

Linkage disequilibrium (LD) decay around SNPs significantly associated with the phenotyped traits was estimated using pairwise Pearson correlations ($r^2$) values calculated from SNP dosage data. Based on whole genome and per chromosome LD decay plots analyses were restricted to a ± 5 Mb window around each focal SNP. Local LD patterns were summarised using LOESS smoothing of $r^2$ as a function of physical distance. The genomic extent of non-negligible LD was defined as the distances on either side of the focal SNP at which LOESS-predicted $r^2$ fell below 0.10. All analyses were performed in R and visualised using ggplot2 [39]. The predicted genes from the 'Malling Jewel' NIAB Genome v1.0 Assembly and Annotation [4] located within the LD decay window of each QTL were downloaded, and annotated genes with a putative role in the traits investigated were identified as candidates by manual scrutiny of the gene lists.

## Results

### Plant material phenotyping

Phenotyping was performed for four traits; plant vigour, leaf colour, fruit size, and thorn density in order to determine the efficacy of the SNP panel developed for genetic investigation in red raspberry. A total of 457 accessions, including 161 F1 progeny of the Paris×486 mapping population, 27 cultivars, and 269 breeding lines were phenotyped. The distribution of phenotypes scored approximated a normal distribution for plant vigour, leaf colour, and fruit size. The distribution of thorn density, however, was highly skewed. Box-Cox transformation, fitted the trait distribution for thorn density to a more normal distribution, and the transformed data were used for subsequent association and QTL analyses (**Fig 1**). The histograms showing the phenotypic distributions for the Paris×486 mapping population are shown in **S1 Fig**. The best linear unbiased estimates (BLUEs) calculated for fruit size, leaf colour, plant vigour, and thorn density in the red raspberry diversity panel and used for subsequent GWAS analysis are given in **S2 File**.

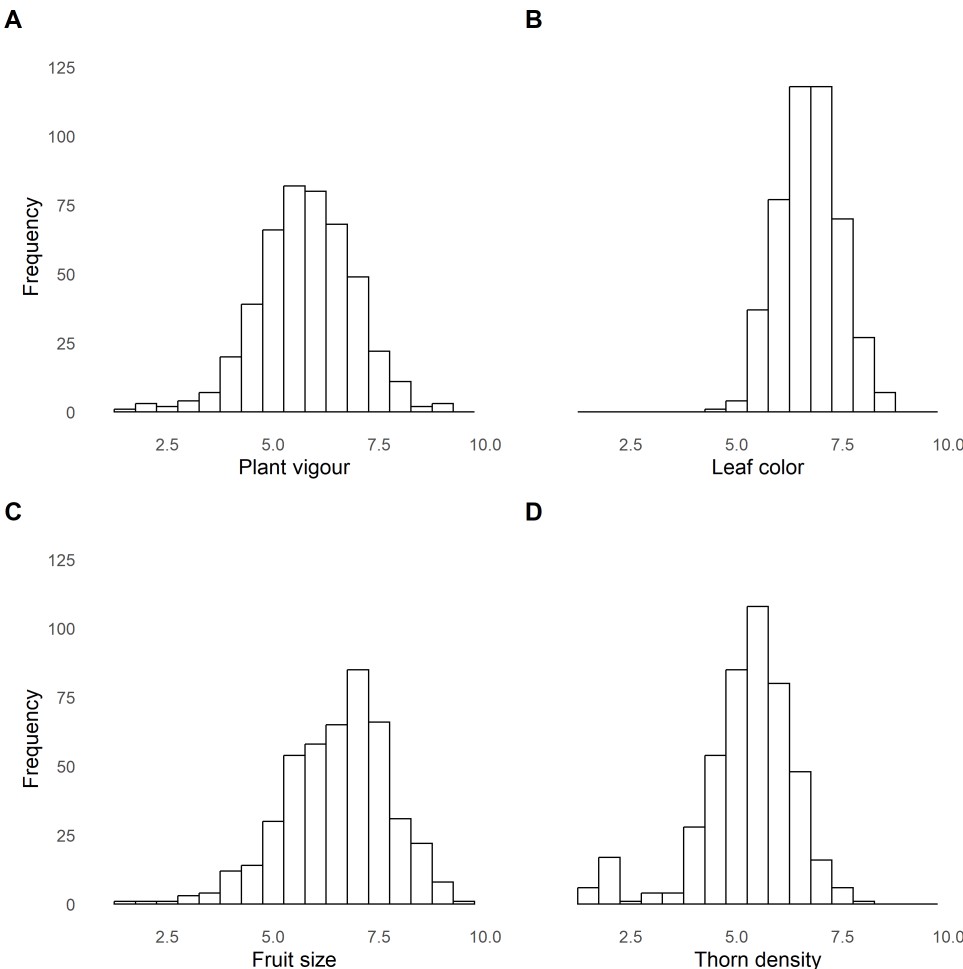

**Fig 1. The distributions of phenotypes in the diversity panel of 457 red raspberry accessions studied in this investigation.** Distributions of phenotypes for (A) plant vigour, (B) leaf colour, (C) fruit size, and thorn density (D).

## SNP panel design

A total of 12,983 SNPs were selected from the three sources described in the materials and methods and submitted to the LGC Genomics design pipeline. The SNPs were located to 7,104 1 kb windows and following removal of loci outside of the percentage GC range, probes were designed for 7,071 loci, which were predicted *in silico* to contain 8,974 SNPs.

## SNP genotyping in red raspberry germplasm

DNA was extracted from leaf material of the 457 accessions studied in this investigation, and following hybridisation of the Flex-Seq probes, library construction and sequencing using the NexSeq 500 Illumina sequencing platform at LGC Genomics (LGC Genomics GmbH, Germany) a total of 816,693,446 mapped 150 bp paired-end reads were retrieved from the 457 samples. The average number of mapped reads per sample was 1,697,907 (min = 98,992, max = 2,378,375) (S3 File). Following SNP calling a variant call format (VCF) file was produced containing a total of 116,731 unfiltered SNPs. The number of SNPs retained at each stage of filtering is shown in Table 1 and the corresponding bed file (S4 File). Following filtering, a total of 5,639 heterozygous bi-allelic SNP sites (S5 File) were identified derived from the 7,071 probes. In addition to the SNPs, 785 indels were identified in the sequencing data between genotypes, but these were not considered in further analyses. Distribution of the 5,639 bi-allelic SNPs on the 'Malling Jewel' genome sequence is shown in Fig 2. The plot revealed eight large gaps of more than 5 Mb in length where no bi-allelic SNPs were identified. At least one large gap was identified on each of the seven chromosomes, and as such may correspond to the centromeric regions of the raspberry genome. The Neighbour Joining tree showing the relationships between the 457 genotypes used in this study is shown in file S2 Fig. The raw reads from the FlexSeq genotyping were deposited in the ArrayExpress repository at EMBL-EBI (www.ebi.ac.uk/arrayexpress) under the accession number **E-MTAB-15650**.

## Paris×486 Linkage map construction

A total of 4,881 SNPs were heterozygous between the parents of the Paris×486 mapping population, 1,098 were heterozygous only in the female parent (AB×AA), 2,084 were heterozygous only in the male parent (AA×AB), with the remaining 1,699 heterozygous in both parental genotypes (AB×AB). A total of 35 progeny returned 'impossible' genotypes following genotyping and were removed from further study, leaving a total of 126 progeny for linkage analysis. Following initial linkage analysis, removal of markers that did not cluster to the expected group based on their physical location on the 'Malling Jewel' genome and subsequent imputation, a total of 4,736 segregating markers were mapped to seven linkage groups representing the seven chromosomes of the red raspberry genome (S3 Fig). Regression

**Table 1. The number of sequenced SNPs retained at each stage of the filtering process in 457 *R. idaeus* selections.**

| Purpose | Call | Remaining markers |
|---|---|---|
| Filter variants with missing genotypes gt 0.1 | bcftools filter -e 'QUAL < 30' gatk1.vcf -O v | 116,731 |
| Filter on read quality | bcftools filter -e 'QUAL < 30' | 116,731 |
| Filter on read depth | bcftools filter -i 'FORMAT/DP > 10' | 116,731 |
| Filter on BaseQRankSum* | bcftools filter -i 'BaseQRankSum < 0.3 && BaseQRankSum> -0.3' | 28,472 |
| Filter on allele frequency | bcftools view -i 'AF > 0.01' | 8,323 |
| Only biallelic snps | bcftools view -v snps --max-alleles 2 | 5,639 |
| Only indels | bcftools view -v indels --max-alleles 2 | 785 |

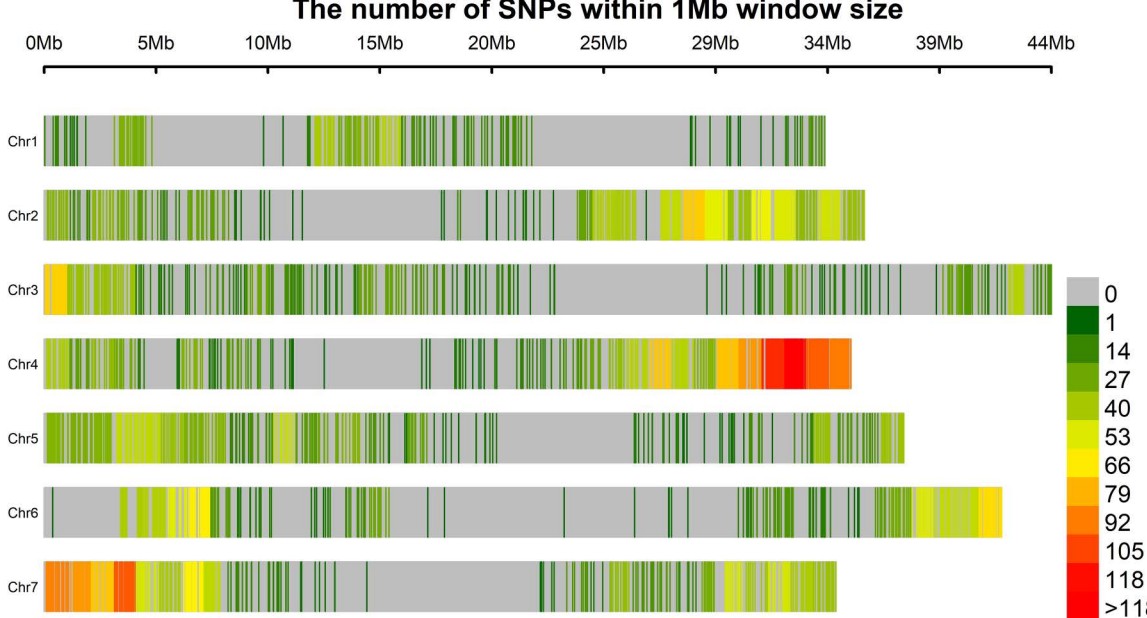

**Fig 2. A heatmap showing the distribution of 5,639 biallelic SNP markers in 1 Mbp windows across the seven chromosomes of the 'Malling Jewel' raspberry reference genome sequence.**

mapping with the selected subset of 590 markers described in the materials and methods produced a male and female linkage map spanning the expected seven chromosome and covering a genetic distance of 610.8 cM and 654.5 cM for the male and female maps respectively. Marker ordering was well-conserved between the male and female maps and collinearity to the 'Malling Jewel' reference genome was high with only one region at the proximal end of chrom2 displaying a putative 'inversion'. This discrepancy was most likely a mis-ordering of markers on the genetic map due supressed recombination, as it preceded a large physical interval containing very few genetic markers (**Fig 3**). Linkage groups were designated following standard nomenclature for *R.idaeus* (from Ri1 to Ri7) with A and B denoting the female 'Paris' and male '486' linkage maps respectively.

**QTL analysis of the four phenotypic traits in the Paris×486 mapping population**

Following QTL analysis and permutation testing, a total of five significant QTL surpassing the LOD threshold of 5.0 were discovered for the four phenotypic traits investigated. One significant QTL for fruit size was identified on the distal end of Ri2, with the most significant marker explaining 17.9% of the phenotypic variation observed, one QTL was identified for plant vigour on the proximal end of Ri3 with the most significant marker explaining 23.7% of the phenotypic variation observed, two QTL were observed for leaf colour on the proximal end of Ri3A and the distal end of Ri5A, with the most significant markers explaining 22.6% and 24.8% of the phenotypic variation observed respectively, and a final QTL for thorn density explaining 32.1% of the phenotypic variation observed was identified on the proximal end of Ri3 (**Fig 4**). All QTL were identified in both the male and female maps and the most significant marker trait associations along with their respective percentage phenotypic variance explained and associated LOD values and the marker positions on the Paris×486 linkage map are given in **Table 2**. Box plots of the most significant markers vs. phenotypes are given in **S4 Fig**.

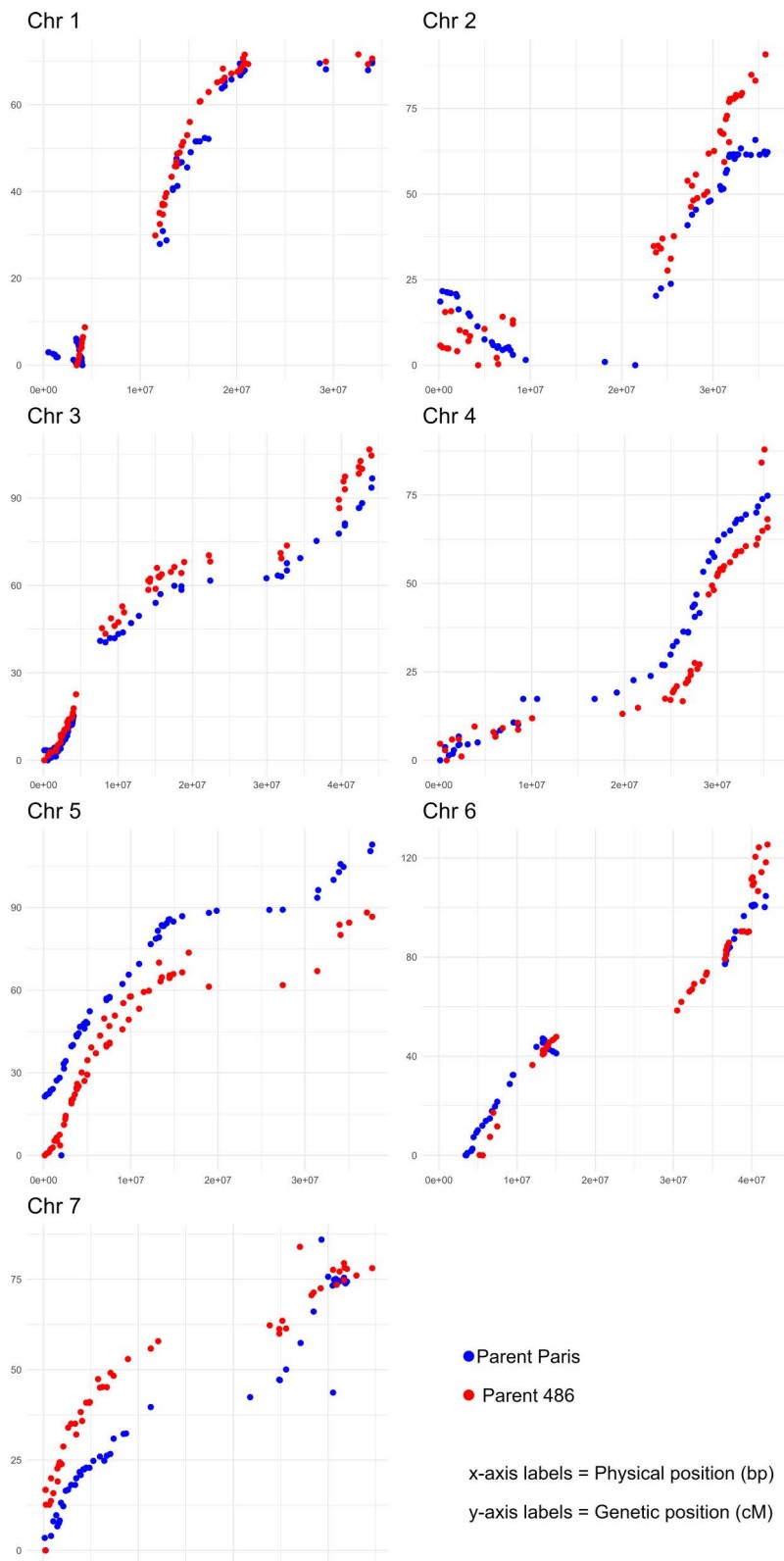

**Fig 3. A plot of the genetic vs. physical positions of the SNP markers mapped in the Paris×486 mapping population.** Markers in blue are those mapped to the 'Paris' (A) map whilst those in red are those mapped to the '486' (B) map.

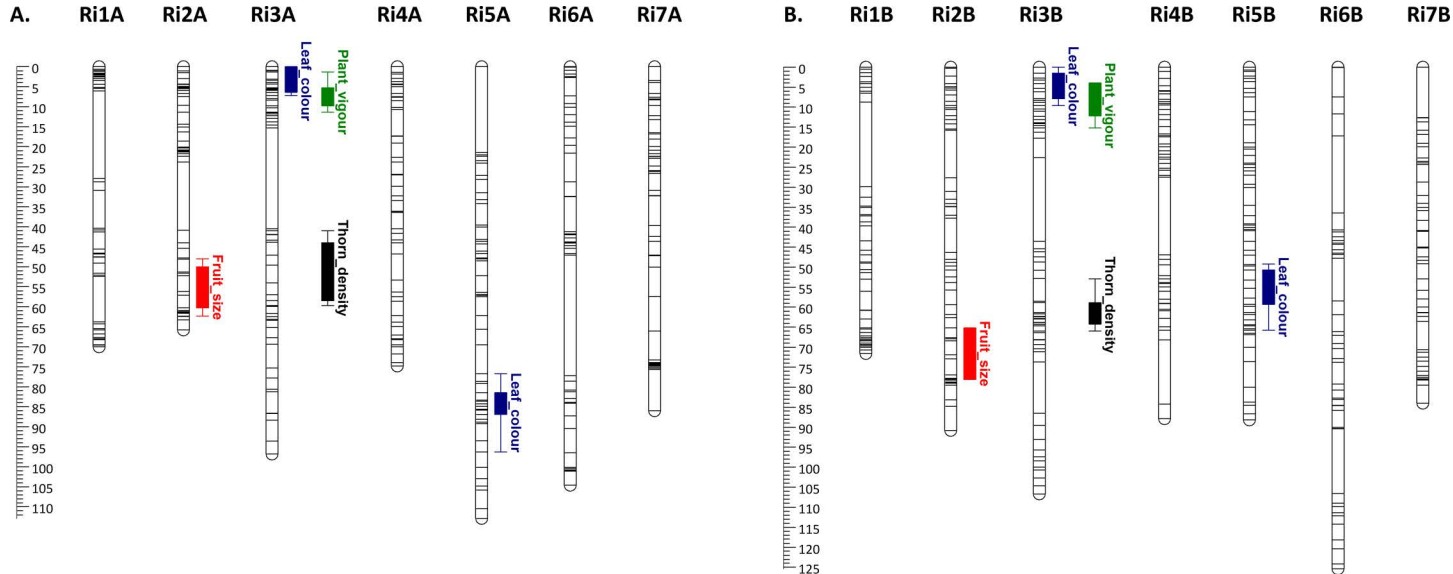

**Fig 4. Genetic linkage map and QTL summary for fruit size, leaf colour, plant vigour and thorn density derived from the raspberry Paris×486 mapping population.** QTL are represented by bars (1-LOD interval) and extended lines (2-LOD interval). Colours served to distinguish between QTL for different traits: red – fruit size; blue – leaf colour; green – plant vigour; black – thorn density. A. represents the linkage map of the female 'Paris' parent, whilst B. represents the male 'S0486' parent. Linkage groups are to scale and represented in cM according to the scale bars to the left of each linkage map.

## GWAS for phenotypic traits in the red raspberry diversity panel

Significant associations were identified for three of the four phenotypic traits analysed using BLINK, implemented in GAPIT: fruit size, leaf colour, and thorn density (**Fig 5**). A highly significant association was observed for thorn density on chromosome 4, with peaks of lower significance on chromosome 5 and 7, a significant association was observed for fruit size on chromosome 1, three significant associations were observed on chromosomes 3, 4 and 5 for leaf colour, whilst no significant associations were identified for plant vigour. The most significant association observed was for thorn density, with a signal peak on chromosome 4 at 33,160,893 bp explaining 34.8% of the phenotypic variation observed. Similar observations were made implementing MLM in GAPIT, however, some of the associations were not as significant using this algorithm (**S5 Fig**) Box plots of the most significant markers vs. phenotypes are given in **S6 Fig**.

## Candidate gene identification

The LD plots of each of the 11 most significant SNPs from the QTL and GWAS analyses were scrutinised and the LD windows in **S7 Fig** were mined from the data of [4] to identify candidate genes. The most significant GWAS association with thorn density was at 33,160, 893 bp in the 'Malling Jewel' reference genome sequence. This marker was found to be in close proximity to the predicted gene jg27953.t1, in LD at position 33,102,286 bp to 33,104,149 bp in the 'Malling Jewel' reference genome sequence that encoded a predicted homeobox-leucine zipper (HOX3) protein (**S6 File**). The most significant QTL for leaf colour were identified on Ri3 and Ri5 in both the mapping population and the GWAS study, with the most significant markers located at 1,652,814 bp and 2,822,294 bp in the mapping study and GWAS respectively on Ri3 and at 9,150,237 bp and 11,495,066 bp in the mapping study (male linkage map) and GWAS respectively. Candidate genes in LD were identified in all four regions associated with anthocyanin biosynthesis. On chromosome 3, predicted gene jg346.t1 was identified, located at 1,450,849 bp to 1,452,587 bp which encodes for a predicted anthocyanidin

**Table 2. The significant QTLs identified in the Paris×486 mapping population, along with the most closely associated SNP markers, their physical position on the 'Malling Jewel' genome sequence, their corresponding genetic position and the LOD and percentage of observed variance explained by each QTL.**

| Trait | Marker | Physical position (bp) | Linkage group | Genetic position (cM) | LOD | Variance explained (%) |
|---|---|---|---|---|---|---|
| Fruit size | c2_31392842 | 31,392,842 | Ri2A/B | 56.199 | 5.31 | 17.7 |
| Fruit size | c2_31520020 | 31,520,020 | Ri2A/B | 57.051 | 5.39 | 17.9 |
| Fruit size | c2_31751824 | 31,751,824 | Ri2A/B | 60.884 | 4.44 | 15 |
| Leaf colour | c3_867579 | 867,579 | Ri3A | 0.966 | 6.52 | 21.2 |
| Leaf colour | c3_1652814 | 1,652,814 | Ri3A | 1.309 | 6.81 | 22 |
| Leaf colour | b3_1691853 | 1,691,853 | Ri3B | 4.059 | 6.97 | 22.6 |
| Leaf colour | c3_1942363 | 1,942,363 | Ri3A | 3.074 | 6.52 | 21.2 |
| Leaf colour | c5_13556169 | 13,556,169 | Ri5A/B | 83.58 | 7.79 | 24.8 |
| Leaf colour | c5_13573690 | 13,573,690 | Ri5A/B | 83.596 | 7.79 | 24.8 |
| Leaf colour | a5_14348691 | 14,348,691 | Ri5A | 85.633 | 7.95 | 25.4 |
| Leaf colour | c5_14454153 | 14,454,153 | Ri5A/B | 85.754 | 7.72 | 24.7 |
| Vigour | b3_2313205 | 2,313,205 | Ri3B | 8.757 | 6.83 | 22.1 |
| Vigour | a3_2556608 | 2,556,608 | Ri3A | 7.204 | 6.85 | 22.2 |
| Vigour | c3_2694951 | 2,694,951 | Ri3A/B | 7.212 | 6.85 | 22.2 |
| Vigour | c3_2822294 | 2,822,294 | Ri3A/B | 8.021 | 7.4 | 23.7 |
| Vigour | a3_2909967 | 2,909,967 | Ri3A | 7.17 | 6.86 | 22.2 |
| Vigour | c3_3090358 | 3,090,358 | Ri3A/B | 8.402 | 6.8 | 22 |
| Thorn density | a3_11758215 | 11,758,215 | Ri3A | 47.102 | 9.39 | 32.1 |
| Thorn density | a3_12805561 | 12,805,561 | Ri3A | 49.564 | 9.51 | 31.9 |
| Thorn density | b3_14248398 | 14,248,398 | Ri3B | 61.322 | 9.57 | 29.9 |
| Thorn density | b3_14248464 | 14,248,464 | Ri3B | 61.289 | 9.63 | 30.1 |
| Thorn density | b3_14306747 | 14,306,747 | Ri3B | 62.258 | 9.57 | 29.6 |
| Thorn density | c3_15046391 | 15,046,391 | Ri3A/B | 54.038 | 9.12 | 28.4 |

reductase gene, whilst four gene predictions jg637.t1, jg638.t1, jg639.t1, and jg640.t1 were located between 2,845,810 bp and 2,854,683 bp on the 'Malling Jewel' reference genome sequence that all encode for predicted anthocyanidin 3-O-glucoside 2''-O-glucosyltransferase-like genes. On chromosome 5, predicted gene jg19747.t1 was located in LD at 7,905,815 bp to 7,907,502 bp in the 'Malling Jewel' reference genome sequence that encodes for anthocyanidin synthase, whilst predicted gene jg20417.t1 was located in LD between 11,965,651 bp and 11,967,018 bp that is predicted to encode an anthocyanidin 3-O-glucosyltransferase 7-like gene.

## Discussion

The main objective of this study was the development of a SNP marker genotyping panel for *R. idaeus* of broad utility to genetics studies in the species. To this end, a total of 5,639 SNP markers heterozygous in a diversity panel of 457 red raspberry genotypes was developed and validated through GWAS, and the development of a genetic linkage map for a bi-parental mapping population. Marker density was similar to other SNP panels that have been developed with analogous technologies for other plant species such as blueberry [21] and comparable to other SNP-based linkage mapping approached in red raspberry such as Genotyping by Sequencing [16,17], but with the advantage of being proven to be highly reproducible when applied to highly heterozygous plant genomes [22]. Marker distribution throughout the *R. idaeus* genome ensured that, whilst some regions contained a higher abundance of heterozygous SNPs, the majority of the coding portion of the genome was represented, whilst marker density was low within the repetitive regions of the genome

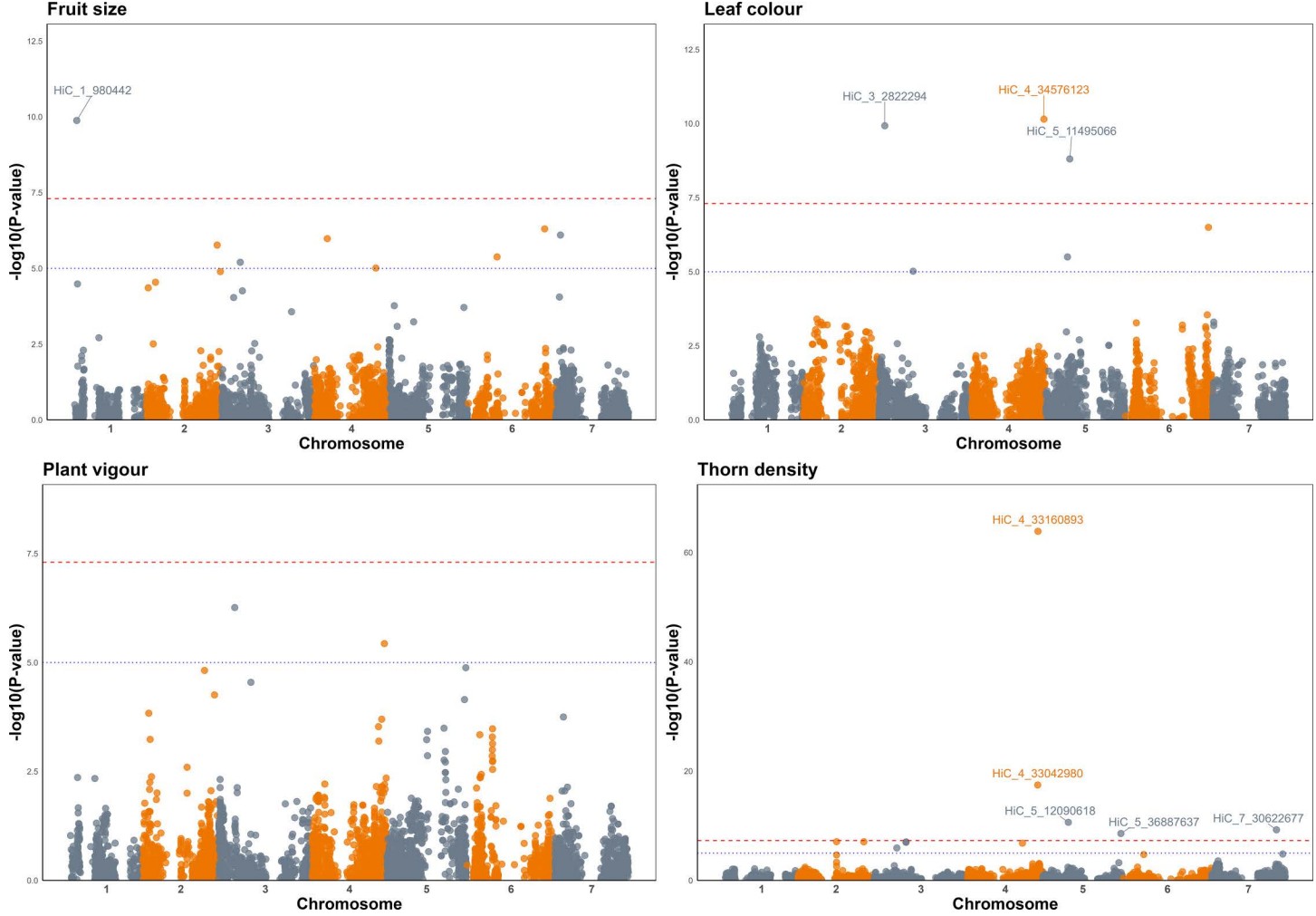

**Fig 5. Manhattan plots of markers significantly associated with fruit size, leaf colour, and thorn density in the red raspberry diversity panel used in this investigation, constructed against the 'Malling Jewel' reference genome sequence using BLINK implemented in GAPIT showing marker-trait associations exceeding the $-\log_{10}(p)$ genome-wide threshold of ~7.5 (red dashed line).** The suggestive threshold of $-\log_{10}(p) \sim 5$ is also indicated with the blue dotted line.

[4]. However, due to the adaptable nature of the FlexSeq genotyping platform, additional SNPs from the highly repetitive regions of the red raspberry genome could be added for future genotyping studies using the marker set presented here.

Using phenotypic data collected over several seasons, association studies revealed significant QTL for several traits in both a diversity panel of red raspberry genotypes and a bi-parental mapping population. The most significant QTL identified was for thorn density, a trait first mapped in red raspberry by Molina-Bravo et al. [44], and in tetraploid blackberry, a close relative of red raspberry by Castro et al. [45], both of whom mapped the trait to linkage group 4 of the respective genomes. In the diversity panel, a single major QTL was identified on chromosome 4 associated with a SNP physically located at 33,160, 893 bp in the 'Malling Jewel' reference genome sequence. Genetic studies performed recently in tetraploid blackberry, identified a genetic region controlling the 'prickle-free' trait in the blackberry cultivar 'Merton Thornless' [46]. This locus was associated with a SNP marker located in an LD block at 33.64 Mbp on the blackberry genome sequence [47] and the authors revealed a potential causal mutation in a homologue of the HOX3 gene located between

33,599,420 and 33,601,244 bp in the blackberry genome, 35 kb from the peak marker. Scrutiny of the gene predictions in the region surrounding the peak SNP for thorn density in this investigation revealed a candidate gene jg27953.t1 located between 33,102,286 bp and 33,104,149 bp on the 'Malling Jewel' reference genome sequence, 4,607 bp from the peak SNP, that encoded a predicted homeobox-leucine zipper (HOX3) gene, suggesting the QTL in red raspberry and black-berry may be controlled by the same gene homologue. Interestingly, the bi-parental mapping population did not reveal any thornless offspring and segregated only for thorn density. The QTL revealed in this population was located on chromo-some 3, between 11–15 Mbp, and as such is controlled by a different gene to that controlling thornlessness on chromo-some 4.

Candidate genes were also discovered in the QTL regions controlling leaf colour on chromosome 3 and chromosome 5 in both the diversity panel and the mapping population. On chromosome 3 in the biparental mapping population pre-dicted gene jg346.t1, located between 1,450,849 bp and 1,452,587 bp, 200,227 bp from the peak SNP is predicted to encode for anthocyanidin reductase. Recently, loss of function mutations in anthocyanin reductase, a key gene in the proanthocyanidin synthesis pathway, have been shown to activate anthocyanin synthesis in strawberry, a close relative of red raspberry, with EMS induced mutants displaying increased anthocyanin accumulation during fruit development and higher anthocyanin content in ripe fruits compared to wild-type clones [48]. On chromosome 3 in the diversity panel four genes jg637.t1, jg638.t1, jg639.t1, and jg640.t1 were located between 2,845,810 bp and 2,854,683 bp, 23,516 bp from the peak SNP, all encoding a predicted anthocyanidin 3-O-glucosyltransferase-like gene. The peak SNP for the significant QTL for leaf colour on chromosome 5 in the diversity panel was located at 11,495,066 bp, 470,585 bp from the predicted gene jg20417.t1 (located between 11,965,651 bp and 11,967,018 bp) encoding an anthocyanidin 3-O-glucosyltransferase 7-like gene. UFGT enzymes are critical for anthocyanin synthesis, acylation, and glucosylation in horticultural plants, as well as in the stabilization of anthocyanin pigments [49], and anthocyanidin-3-O-glucosyltransferases (UFGTs) have been shown to play a key role in the biosynthesis of stable anthocyanins, with anthocyanin accumulation significantly reduced in UFGTs mutants in *Arabidopsis* [50]. The peak SNP for the significant QTL for leaf colour on chromosome 5 was located at 9,150,237 bp in the mapping study, whilst gene jg19747.t1, encoding a predicted anthocyanidin synthase was located between 7,905,815 bp and 7,907,502 bp, 1,244,422 bp from the peak SNP in the 'Malling Jewel' reference genome sequence. Recently, mutations in anthocyanidin synthase were implicated in the loss of fruit colour in the red raspberry mutants 'Anne' [51] and 'Varnes' [3].

Whilst no causal mutations were identified in this study through linkage mapping or GWAS, the results demon-strate the efficacy of the SNPs in identifying QTL for agronomic traits and this the potential utility of the SNP panel developed for studies into the genetic basis of other important traits in red raspberry. It also demonstrates the poten-tial for the SNP panel for use in the development of tools for marker assisted breeding, and for genomic prediction and selection, as has been shown in other species such as strawberry [52,53] and chickpea (*Cicer arietinum* L.) [54], especially for complex polygenic traits such as yield and disease resistance which are of significant importance to commercial raspberry cultivation.

## Conclusions

The SNP panel presented here has been shown to effective at developing linkage maps, and detecting QTL for traits of agronomic importance in red raspberry in both a diversity panel and a bi-parental mapping population. As such, the panel could assist in significantly accelerating the development of new raspberry cultivars with improved traits, such as enhanced disease resistance, better fruit quality, and increased yield, through the application of SNP markers to breeding and selection. It will also be of utility for characterising and conserving genetic diversity, as well as for safeguarding the genetic security of commercial raspberry varieties through genetic fingerprinting, and as such should prove to be a valu-able resource for researchers and breeders in the global raspberry community.

## Supporting information

**S1 File. The 27 red raspberry varieties included in the diversity panel in this investigation.**
(DOCX)

**S2 File. Best linear unbiased estimates (BLUEs) calculated for fruit size, leaf colour, plant vigour, and thorn density from the red raspberry diversity panel used for GWAS in this investigation.**
(XLSX)

**S3 File. Summary of sequencing reads, including the number of adapter-clipped reads received from the sequencing facility (LGC Genomics GmbH, Germany), the number of reads retained after Trimmomatic trimming, and the number of reads successfully mapped to the Malling Jewel genome.**
(DOCX)

**S4 File. The BED file corresponding to the positions of the 5,369 SNP markers used in the current investigation.**
(TXT)

**S5 File. The numeric genotypes used in the current investigation.** A '0' indicates homozygous reference, a '1' indicates a heterozygous, and a '2' a homozygous alternate.
(TXT)

**S6 File. Predicted genes from the 'Malling Jewel' reference genome sequence within QTL regions spanning 2 Mb intervals.** Each sheet corresponds to one of the five studied traits, reporting for each gene the protein ID, genomic position, and best BlastP hit against NCBI-nr, Araport11, RefSeq, SwissProt, and TrEMBL databases.
(XLSX)

**S7 File. Statement of Inclusivity in global research.**
(DOCX)

**S1 Fig. The distributions of phenotypes in the Paris×486 mapping population.** Distributions of phenotypes for (A) plant vigour, (B) leaf colour, (C) fruit size, and (D) thorn density.
(TIFF)

**S2 Fig. A Neighbour Joining Tree showing the relationships between the 457 red raspberry genotypes used in this study using data from the 5,369 bi-allelic SNP markers developed here.**
(PDF)

**S3 Fig. Maximum Likelihood linkage map for the 4,736 segregating markers that located to the expected chromosome according to physical position in the Paris×486 mapping population.** Markers segregating AB×AB that were mapped to both male and female maps are linked by a solid line between linkage maps.
(PNG)

**S4 Fig. Box Plots showing the distribution of the fruit size, leaf colour, plant vigour, and thorn density phenotypes between homozygous and heterozygous genotypes of the most significant SNPs identified in the QTL analysis presented here.**
(PNG)

**S5 Fig. Manhattan plots of markers significantly associated with fruit size, leaf colour, and thorn density in the red raspberry diversity panel used in this investigation, constructed against the 'Malling Jewel' reference genome sequence using MLM implemented in GAPIT showing marker-trait associations exceeding the $-\log_{10}(p)$**

**genome-wide threshold of ~7.5 (red dashed line).** The suggestive threshold of -log$_{10}$(p) ~ 5 is also indicated with the blue dotted line.
(TIFF)

**S6 Fig. Box Plots showing the distribution of the fruit size, leaf colour, plant vigour, and thorn density phenotypes between homozygous and heterozygous genotypes of the most significant SNPs identified in the GWAS analysis presented here.**
(PNG)

**S7 Fig. Local linkage disequilibrium (LD) around trait-associated focal SNPs.** Pairwise LD ($r^2$) was calculated between each focal SNP and surrounding markers within a ± 5 Mb window and plotted against physical distance. LOESS-smoothed curves summarise local LD decay, with vertical dashed lines indicating the focal SNP and the positions where LOESS-predicted $r^2$ drops below 0.10 (solid lines).
(TIF)

## Author contributions

**Conceptualization:** Jahn Davik, DJ Sargent.

**Data curation:** Jahn Davik, Paolo Zucchi, L Milne, J Graham, DJ Sargent.

**Formal analysis:** Jahn Davik, Matteo Buti, DJ Sargent.

**Investigation:** Jahn Davik, DJ Sargent.

**Resources:** Paolo Zucchi, L Milne, J Graham.

**Validation:** Paolo Zucchi, Matteo Buti.

**Writing – original draft:** Jahn Davik, DJ Sargent.

**Writing – review & editing:** Jahn Davik, Paolo Zucchi, Matteo Buti, L Milne, J Graham, DJ Sargent.

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
