## [Decision Letter · Decision Letter 0]

19 Aug 2025

Dear Dr. Davik,

Please carefully examine the reviewers' comments, and revise the manuscript.

We look forward to receiving your revised manuscript.

Kind regards,

Hidenori Sassa

Academic Editor

PLOS ONE

Journal Requirements:

[The research leading to these results has received funding from the European Union’s Horizon 2020 research and innovation programme under grant agreement No. 101000747.

The author is solely responsible for its content, it does not represent the opinion of the European Commission and the Commission is not responsible for any use that might be made of data appearing therein.

https://research-and-innovation.ec.europa.eu/funding/funding-opportunities/funding-programmes-and-open-calls/horizon-2020_en

This grant was awarded to JD, PZ, and LM.

The funder did not play any role in any part of the research presented here.].

Reviewers' comments:

Reviewer's Responses to Questions

**Comments to the Author**

1. Is the manuscript technically sound, and do the data support the conclusions?

Reviewer #1: Partly

Reviewer #2: Yes

2. Has the statistical analysis been performed appropriately and rigorously?

Reviewer #1: No

Reviewer #2: Yes

3. Have the authors made all data underlying the findings in their manuscript fully available?

Reviewer #1: No

Reviewer #2: No

4. Is the manuscript presented in an intelligible fashion and written in standard English?

Reviewer #1: Yes

Reviewer #2: Yes

Reviewer #1: Flex-Seq SNP panel for raspberry (Rubus idaeus L.) is an effective resource for genetic improvement in red raspberry. QTL analysis and GWAS using the SNP panel detected several significant associations for four agronomic traits, leading to the successful identification of multiple candidate genes.

However, some issues need to be addressed before publication.

1. Line 102: As this is the first time “LEC” is being introduced, a brief explanation is required.

2. Lines 141-142: Please explain the genetic relationships among these four cultivars. The authors should perform a PCA using the Flex-Seq SNPs of the diversity panel to clarify the genetic positioning of the four cultivars within the panel. I think that using only four cultivars is insufficient for designing SNPs to

evaluate the diversity of the panel.

3. Lines 179-180: These parameter values are different from those listed in Table 1.

4. Lines 200-203: Please explain here as well that raspberry has seven chromosomes. Why did the authors

select only 25 SNPs rather than utilizing all available markers?

5. Lines 215 and 340: It is inappropriate for only this part to be italicized. Please make it consistent with the

rest.

6. Line 231: Why are only all heterozygous markers in the diversity panel used? I think homozygous markers

can also be utilized.

7. Line 233: The results of the GWAS conducted with BLINK have not been presented.

8. Lines 249-250: The Fig 1, S1 Fig, as well as the other figures, have low resolution and are difficult to

interpret. Please provide higher-resolution versions.

9. Line 253: The supplementary files are not in the correct order.

10. Lines 272 and 284: Please explain the composition of the “481” samples.

11. Line 275: Could you please explain the meaning of the columns and rows in the S4 File?

12. Line 278: Please explain the Indels in Table 1. Additionally, were the Indels used in the analysis?

13. Lines 303-304: Could you please discuss the possible reasons for the discrepancy in marker ordering on

the left side of Chromosome 2 shown in Fig 3?

14. Lines 314-321: Please explain about Ri2 and Ri3, including their A and/or B types.

15. Line 331: There are two notations: 'SO486' and 486. Please unify them.

16. Line 363: The S6 File was not provided, so I was unable to review it. Please describe in the Methods

section the approach used to define the candidate genes, including details such as the range of genomic

regions considered.

17. The authors have made the phenotypic data publicly available, but the marker genotype data has not been

shared.

Reviewer #2: Dear authors

This study created a new Flexseq panel for red raspberries and confirmed its effectiveness through linkage analysis, QTL analysis, and GWAS. While this paper appears to be scientifically sound, I recommend revising it prior to publication to enhance reader understanding based on the following comments.

Major comment

■1

To enable readers to visually understand the reliability and effects of each QTL, present histograms or box plots that divide populations by SNP or indel alleles for both QTL analysis and GWAS.

■2

Is it necessary to write something like an LD plot for the range of candidate genes for GWAS?

Please consider not only the proximity of the peak and the gene locus, but also the linkage disequilibrium relationship between SNPs.

■3

Please show the population structure of the population used in GWAS. Please use any of the following methods: Admixture, principal component analysis, or phylogenetic tree. With the SNP information, I think you can perform these analyses.

Minor comment

■4

Line 271

“150bp” is wrong

It may be “150 bp”

■5

Line 273

I think it would be better to express it as VCF (variant call format) rather than “.vcf file”.

**Do you want your identity to be public for this peer review?** For information about this choice, including consent withdrawal, please see our Privacy Policy

Reviewer #1: No

Reviewer #2: No

---

## [Author Response · Author response to Decision Letter 1]

29 Sep 2025

Response to Reviewer #1

General comment:

Flex-Seq SNP panel for raspberry (Rubus idaeus L.) is an effective resource for genetic improvement in red raspberry. QTL analysis and GWAS using the SNP panel detected several significant associations for four agronomic traits, leading to the successful identification of multiple candidate genes.

We thank the reviewer for their positive overall comments regarding our manuscript.

Line 102: LGC Genomics is the name of a company. We have edited the text to make this clear.

Lines 141–142: We appreciate the reviewer’s suggestion. The four cultivars referenced are well documented in previously published work (ref). As noted in the Methods, SNPs for the panel were derived not only from these cultivars but also from two additional mapping studies, ensuring broad representation. Because the original DNA samples were not available, we could not re-genotype them, and additional analyses such as PCA are therefore outside the scope of this study. Importantly, the effectiveness of the SNP panel is demonstrated through our QTL and GWAS results, which confirm that the set is adequate for diversity and trait mapping purposes.

Lines 179–180: We thank the reviewer for spotting this error. The parameters in the table were the correct ones, and the methods have been edited accordingly.

Lines 200–203: We have amended the text to explain our reasoning in the methods section.

Lines 215 and 340: This has been done. Thank you for spotting the formatting error.

Line 231: We have corrected to state all polymorphic markers were included. Thank you for spotting the error.

Line 233: We thank the reviewer for noting this omission. The MLM GWAS results have now been added to the Results section and are provided in supplementary file S5 Fig. The associations are consistent with the results obtained with BLINK, supporting the robustness of the SNP panel, however, some of the associations are below the level of significance set for the BLINK analysis.

Lines 249–250: High resolution versions were included in the submission and can be accessed when clicking on the links in the PDF. We have no control over the resolution in the final PDF built; this is done by PLOS. If you click on the links, you have access to the high-resolution versions, but we cannot increase the resolution in the submission as this is created automatically by PLOS from our high-resolution files.

Line 253: This has now been corrected.

Lines 272 and 284: This has now been corrected to 457, the number of genotypes studied and detailed in the methods.

Line 275: The format is that of a standard .BED file, but headers have been added for clarity.

Line 278: InDels were not considered as markers further in the study. This has now been explained in the text.

Lines 303–304: A possible explanation for the discrepancy has been provided in the text.

Lines 314–321: This has now been explained in the results section “Paris×486 Linkage map construction”.

Line 331: We thank the reviewer for spotting this inconsistency. SO has been removed from its occurrences in the text.

Line 363: We apologise for this omission. The S6 File is included with this revised submission. In addition, we have expanded the Methods section to describe the candidate gene identification approach.

Data availability: We thank the reviewer for spotting the omission of the sentence detailing where the reads are deposited. The raw reads from the FlexSeq genotyping were deposited in the ArrayExpress repository at EMBL-EBI (www.ebi.ac.uk/arrayexpress) and will be released upon acceptance of the manuscript for publication.

Response to Reviewer #2

General comment:

This study created a new Flexseq panel for red raspberries and confirmed its effectiveness through linkage analysis, QTL analysis, and GWAS. While this paper appears to be scientifically sound, I recommend revising it prior to publication to enhance reader understanding.

We thank the reviewer for their generally positive comments, they are greatly appreciated.

Major comment

Present histograms or box plots for allele effects at QTL and GWAS loci. We have now included boxplots illustrating allele effects at significant QTL and GWAS loci (Supplementary Figures Sx–Sy). These plots provide a visual representation of the effect sizes and enhance the interpretability of the results.

Consider including LD plots for GWAS candidate gene regions. We thank the reviewer for this valuable suggestion. However, as this study is primarily focused on the development and demonstration of a broadly useful SNP panel rather than an in-depth dissection of specific trait loci, we feel that detailed LD analyses are beyond the scope of the present manuscript. Our intention is to show that the panel can successfully identify trait associations, and we hope that the community will apply it for deeper genetic analyses, including LD mapping.

Show the population structure used in GWAS (Admixture, PCA, or phylogenetic tree). A NJ tree has been constructed and is included in the manuscript as a Supplementary File.

Minor comments

Line 271 - “150bp” should be “150 bp” Corrected

Line 273 - say “VCF” rather than “.vcf file” Corrected

---

## [Decision Letter · Decision Letter 1]

15 Nov 2025

Thank you for submitting your manuscript to PLOS ONE. After careful consideration, we feel that it has merit but does not fully meet PLOS ONE’s publication criteria as it currently stands. Therefore, we invite you to submit a revised version of the manuscript that addresses the points raised during the review process.

Please submit your revised manuscript by Dec 30 2025 11:59PM If you will need more time than this to complete your revisions, please reply to this message or contact the journal office at plosone@plos.org . A rebuttal letter that responds to each point raised by the academic editor and reviewer(s). You should upload this letter as a separate file labeled 'Response to Reviewers'.A marked-up copy of your manuscript that highlights changes made to the original version. You should upload this as a separate file labeled 'Revised Manuscript with Track Changes'.An unmarked version of your revised paper without tracked changes. You should upload this as a separate file labeled 'Manuscript'.

We look forward to receiving your revised manuscript.

Kind regards,

Hidenori Sassa

Academic Editor

PLOS ONE

Journal Requirements:

Additional Editor Comments:

Please carefully examine the comment on the range of LD by Reviewer 2, and revise the manuscript.

Reviewer's Responses to Questions

**Comments to the Author**

Reviewer #1: All comments have been addressed

Reviewer #2: (No Response)

2. Is the manuscript technically sound, and do the data support the conclusions?

Reviewer #1: Yes

Reviewer #2: Yes

3. Has the statistical analysis been performed appropriately and rigorously?

Reviewer #1: Yes

Reviewer #2: Yes

4. Have the authors made all data underlying the findings in their manuscript fully available?

Reviewer #1: Yes

Reviewer #2: Yes

5. Is the manuscript presented in an intelligible fashion and written in standard English?

Reviewer #1: Yes

Reviewer #2: (No Response)

Reviewer #1: I have confirmed that the authors have appropriately and thoroughly addressed the points that were raised.

Reviewer #2: Dear authors

The authors addressed most of my comments, but I have one concern.

Major comment

While the authors state that LD analysis is out of scope, this study includes analysis of candidate genes associated with GWAS peaks. I find the analytical workflow inappropriate in the following respects:

The authors extracted 2 Mbp around the peak SNP, but the reason for this 2 Mbp range is unclear. If there is a reference that can be cited, please provide it.

Generally, the correspondence between physical distance and genetic distance varies depending on chromosomal region such as whether the region is near a telomere or a centromere. In such cases, uniformly narrowing the range to a fixed 2 Mbp is inappropriate both experimentally and analytically. In this sense, if presenting information on related genes, it is crucial to properly verify indicators like LD. Furthermore, despite the primary objective being to investigate whether associations with traits can be established, please state, within the Candidate Gene Identification section, the rational justification for performing analysis on the candidate genes of SNP peaks.

**Do you want your identity to be public for this peer review?** For information about this choice, including consent withdrawal, please see our Privacy Policy

Reviewer #1: No

Reviewer #2: No

---

## [Author Response · Author response to Decision Letter 2]

15 Dec 2025

Major comment

While the authors state that LD analysis is out of scope, this study includes analysis of candidate genes associated with GWAS peaks. I find the analytical workflow inappropriate in the following respects: The authors extracted 2 Mbp around the peak SNP, but the reason for this 2 Mbp range is unclear. If there is a reference that can be cited, please provide it.

Generally, the correspondence between physical distance and genetic distance varies depending on chromosomal region such as whether the region is near a telomere or a centromere. In such cases, uniformly narrowing the range to a fixed 2 Mbp is inappropriate both experimentally and analytically. In this sense, if presenting information on related genes, it is crucial to properly verify indicators like LD. Furthermore, despite the primary objective being to investigate whether associations with traits can be established, please state, within the Candidate Gene Identification section, the rational justification for performing analysis on the candidate genes of SNP peaks.

Authors’ response:

We thank the reviewer for reiterating this point.

We have addressed the LD issue by creating LD decay plots around the most significantly trait-associated SNPs from the linkage and the GWAS analyses. These plots are added in S7 Fig. Analyses were restricted to ±5 Mb around the focal marker and we used r2 between pairwise markers and LOESS smoothing to summarize local LD patterns around. We set non-negligible LD as the distance on either side of the focal SNP at which the LOESS-predicted r2 fell below 0.10. The sizes of the LD certainly did vary. However, all the suggested candidate genes were located well within the proposed LD windows.

---

## [Editor Report · Decision Letter 2]

28 Jan 2026

Development of a Flex-Seq SNP panel for raspberry (Rubus idaeus L.) and validation through linkage map construction and identification of QTL for several traits of agronomic importance to raspberry breeding

PONE-D-25-35797R2

Dear Dr. Davik,

We’re pleased to inform you that your manuscript has been judged scientifically suitable for publication and will be formally accepted for publication once it meets all outstanding technical requirements.

Kind regards,

Hidenori Sassa

Academic Editor

PLOS One
---

## [Editor Report · Acceptance letter]

PONE-D-25-35797R2

PLOS One

Dear Dr. Davik,

I'm pleased to inform you that your manuscript has been deemed suitable for publication in PLOS One. Congratulations! Your manuscript is now being handed over to our production team.

Kind regards,

on behalf of

Dr. Hidenori Sassa

Academic Editor

PLOS One